# SparseFormer: Sparse Visual Recognition via Limited Latent Tokens

**Ziteng Gao**
Show Lab, National University of Singapore

**Zhan Tong**
Ant Group

**Limin Wang**
Nanjing University

**Mike Zheng Shou**[✉]
Show Lab, National University of Singapore

## Abstract

Human visual recognition is a *sparse* process, where only a few salient visual cues are attended to rather than every detail being traversed uniformly. However, most current vision networks follow a *dense* paradigm, processing every single visual unit (such as pixels or patches) in a uniform manner. In this paper, we challenge this dense convention and present a new vision transformer, coined *SparseFormer*, to explicitly imitate human's sparse visual recognition in an end-to-end manner. SparseFormer learns to represent images using a highly limited number of tokens (*e.g.*, down to 9) in the latent space with sparse feature sampling procedure instead of processing dense units in the original image space. Therefore, SparseFormer circumvents most of dense operations on the image space and has much lower computational costs. Experiments on the ImageNet-1K classification show that SparseFormer delivers performance on par with canonical or well-established models while offering more favorable accuracy-throughput trade-off. Moreover, the design of our network can be easily extended to the video classification task with promising performance with lower compute. We hope our work can provide an alternative way for visual modeling and inspire further research on sparse vision architectures. Code and weights are available at https://github.com/showlab/sparseformer.

## 1 Introduction

Designing neural architectures for visual recognition has long been an appealing yet challenging topic in the computer vision community. In past few years, there has been an paradigm shift for vision architectures from convolutional neural networks (CNNs) (Krizhevsky et al., 2012; He et al., 2016; Huang et al., 2017; Radosavovic et al., 2020; Liu et al., 2022) to vision transformers (ViTs) (Dosovitskiy et al., 2021; Touvron et al., 2021; Liu et al., 2021a; Wang et al., 2021). Though their underlying operations may vary, both architectures involve the traversal of every pixel or patch in an image to densely model visual patterns and image semantics. This traversal convention stems from the sliding window approaches (Ojala et al., 1996), assuming that foreground objects may appear uniformly at any spatial location in an image, and continues to dominate the design of modern visual neural networks.

However, as humans, we do not need to examine every detail in a scene for perception. Instead, our eyes can quickly identify regions of interest with just few glimpses, and then recognize edges and textures, as well as high-level semantics (Desimone & Duncan, 1995; Marr, 2010; Itti & Koch, 2001; Rensink, 2000). This contrasts greatly with existing vision architectures, which need to exhaustively traverse tons of visual units in the image grid. Such dense convention incurs redundant computation for backgrounds, particularly when scaling up the input resolution. Also, this paradigm cannot directly provide interpretable insights about the specific regions of interest that vision model are focusing on within an image.

---

✉: Corresponding Author.

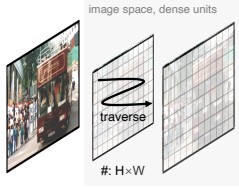 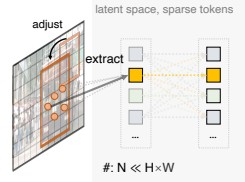

**dense** paradigm          our **sparse** paradigm

Figure 1: **Dense** versus our proposed **sparse** paradigm for visual recognition. The dense paradigm requires traversing $H \times W$ units to perform convolution or attention, while our proposed sparse paradigm performs transformers over only $N$ latent tokens where $N \ll H \times W$.

In this paper, we propose a new vision network architecture, coined **SparseFormer**, to perform *sparse visual recognition* by explicitly imitating the human vision system. SparseFormer learns to represent an image using transformers with a limited number of tokens in the latent space *right from the beginning*. Each latent token is associated with a region of interest (RoI) descriptor, and the token RoI can be adjusted iteratively to focus on foreground regions. With this design, we can highly reduce the number of tokens needed in the transformer (*e.g.*, down to 9) and the overall computational cost of SparseFormer is *almost irrelevant* to the input resolution when a fixed number of latent tokens adopted. Therefore, it is reasonable to refer to our method as a *sparse* approach for visual recognition. Moreover, the RoI adjusting mechanism can be supervised *solely with classification signals* in an end-to-end manner, without prior training with localizing signals.

SparseFormer, as an initial step towards sparse visual recognition, aims to provide an alternative paradigm for vision modeling, rather than achieving state-of-the-art results with bells or whistles. Nonetheless, it still delivers promising results on the challenging ImageNet classification benchmark on par with dense counterparts, but at lower computational costs. Since all transformer operations are performed on a reduced number of tokens in the latent space, SparseFormer has a lower memory footprint and higher throughput compared to dense architectures, especially in the low-compute region. Also, SparseFormer can deliver promising results compatibly with versatile token number settings, *e.g.*, 74.5% on IN-1K with only 9 tokens, and 81.9% with 81 tokens for a tiny variant. Visualizations show SparseFormer can distinguish foregrounds from backgrounds when trained in an end-to-end manner using only classification labels. We also explore several scaling-up strategies of SparseFormer. The simple design of SparseFormer can be further extended to video understanding with minor efforts, where video inputs are more compute-expensive for dense models but well-suited for SparseFormer. Experimental results also demonstrate that our video extension of SparseFormer yields promising performance on the Kinetics-400 with low compute.

## 2 RELATED WORK

**Dense vision architectures.** Since the pioneering AlexNet (Krizhevsky et al., 2012), convolutional neural networks (CNNs) has dominated the field of the visual recognition with comprehensive advancements (He et al., 2016; Huang et al., 2017; Radosavovic et al., 2020; Tan & Le, 2019; Liu et al., 2022; Szegedy et al., 2016; 2015; Ioffe & Szegedy, 2015; Wu & He, 2020). Recently, vision transformers Dosovitskiy et al. (2021) have delivered great modeling capabilities and remarkable performance with the transformer architecture, which originates from the machine translation community (Vaswani et al., 2017). Several variants of vision transformers (ViTs) have been proposed to improve its efficiency and effectiveness with specific designs for vision domain (Liu et al., 2021a; Yuan et al., 2021; Wang et al., 2021; Yue et al., 2021; Fan et al., 2021; Gu et al., 2021; Xia et al., 2022; Tu et al., 2022; Ma et al., 2023; Chang et al., 2023b). However, both of these two architectures require traversing every pixel or patch over an image to perform dense operations, either convolution or attention, in order to obtain the final representation, incurring unnecessary compute burden for backgrounds. This issue becomes more severe when handling data-intensive inputs like videos, as the computation cost increases due to the larger number of units that need to compute (Tran et al., 2015; Carreira & Zisserman, 2017; Tran et al., 2018; Qiu et al., 2017; Xie et al., 2018; Arnab et al., 2021), even when sampling strategies adopted (Wang et al., 2019; Feichtenhofer et al., 2019). To speed up vision transformers, researchers have proposed to gradually reduce vision tokens (Liang et al., 2022; Fayyaz et al., 2022; Bolya et al., 2022; Rao et al., 2021; Yin et al., 2022; Chang et al.,

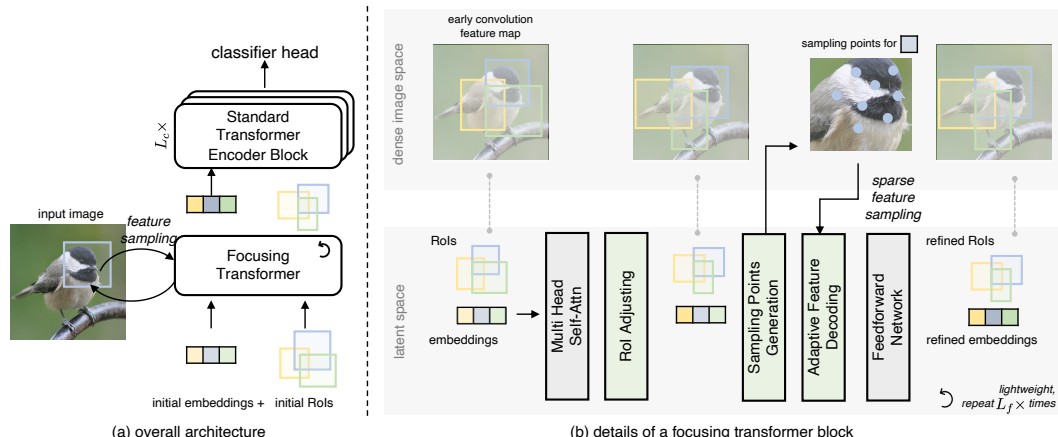

Figure 2: The SparseFormer overall architecture and details of the focusing transformer. Here we depict a set of 3 latent tokens for simplicity. $L_c \times$ standard vision transformer encoder block makes a cortex transformer. Compared with the vanilla transformer block, there are extra RoI adjusting, sampling points generation, and feature sampling & decoding process, repeating itself $L_f \times$ times in a focusing transformer block. All operations in the focusing and cortex transformer are performed over latent token set, except for feature sampling in the image space.

2023a) or exploit sparse attention patterns (Zhang et al., 2021). Different from these methods, SparseFormer directly learns to represent an image via a limited number of tokens right from the beginning in the latent space.

**Perceiver architectures and detection transformers.** Moving vision tokens into the latent space is not a new thing. Perceiver architectures (Jaegle et al., 2021; 2022) for vision learns a set of tokens in the latent space to represent an image via the standard latent transformer decoders with cross attentions. Our proposed SparseFormer is greatly inspired by this paradigm, where the "cross attention" between the image space and the latent space is replaced by the proposed sparse feature sampling. It is worth noting that the standard cross attention also requires traversing every unit over the image feature, while our sparse feature sampling directly extracts features by bilinear interpolation, whose computational complexity is independent of the input resolution. Moreover, Perceiver architectures (Jaegle et al., 2021; 2022) utilize 512 latent tokens, whereas our SparseFormer achieves superior results by using about $0.1 \times$ latent tokens. Detection transformer (DETR) and its variants (Carion et al., 2020; Zhu et al., 2020; Sun et al., 2021) also use latent tokens (queries) to represent objects, which are decoded usually from features extracted by deep convolutional networks and multiple transformer encoder blocks. In contrast, SparseFormer, as a sparse architecture, do not require this heavy image feature design, and instead delievers promising results with a simple early convolution.

**Glimpse models**, proposed in the early days of neural vision research (Mnih et al., 2014; Ranzato, 2014), aimed to imitate explicitly the human visual perception process by capturing selective regions of an image for recognition. While these models are efficient as they just involve limited parts of an image, they often lack differentiability in terms of selecting such regions, and require workarounds like the expectation-maximization algorithm (Dempster et al., 1977) or reinforcement learning to optimize. Our presented SparseFormer architecture can be seen as an multi-glimpse extension model, similiar to (Tan et al., 2021). The whole SparseFormer pipeline can be optimized in an end-to-end manner thanks to the bilinear interpolation. More importantly, SparseFormer proves to be effective on large benchmarks like ImageNet-1K (Deng et al., 2009).

## 3 SPARSEFORMER

In this section, we describe the SparseFormer architecture in detail. Since SparseFormer performs transformer operations over tokens in the latent space, we start with the definition of latent tokens.

**Latent tokens.** Different from dense models, which involve per-pixel or per-patch modeling in the original image space, SparseFormer learns to understand an image via sparse tokens in the latent

space. Similar to queries or latents in transformer decoder architectures (Vaswani et al., 2017; Carion et al., 2020; Jaegle et al., 2021), a latent token in SparseFormer has a corresponding embedding $\mathbf{t} \in \mathbb{R}^d$ in the latent space. To explicitly model spatial focal regions, we associate every latent token $\mathbf{t}$ in SparseFormer with an RoI descriptor $\mathbf{b} = (x, y, w, h)$, where $x, y, w, h$ are center coordinates, width, and height, normalized in the range $[0, 1]$. Therefore, a latent token is made up of an embedding $\mathbf{t}$ representing its content, and an RoI descriptor $\mathbf{b}$ representing its geometric property. Then the whole latent token set of SparseFormer is then described as

$$\mathbf{T} = \{(\mathbf{t}_1, \mathbf{b}_1), (\mathbf{t}_2, \mathbf{b}_2), \cdots, (\mathbf{t}_N, \mathbf{b}_N)\}, \tag{1}$$

where $N$ is the number of latent tokens, both embedding $\mathbf{t}$ and $\mathbf{b}$ can be refined by latent transformers. The initial $\mathbf{t}$ and $\mathbf{b}$ are learnable parameters of SparseFormer, and their initialization will be described in the experiment section and the appendix.

## 3.1 BUILDING LATENT TOKENS

SparseFormer uses two types of transformers in the latent space: the focusing transformer and the cortex transformer, as depicted in Figure 2. The focusing transformer extracts regional image features into token embeddings and adjusts token RoIs iteratively. The subsequent cortex transformer, which is of the standard transformer encoder architecture, takes only latent tokens embeddings as inputs. We first introduce operations that makes a focusing transformer.

**Sparse feature sampling.** In the feature sampling procedure, a latent token first generates $P$ sampling points in the image space based on its geometric property, *i.e.*, its RoI. These sampling points are represented as image coordinates (*i.e.*, $x$ and $y$). Then, the focusing transformer uses bilinear interpolation on these sampling points to extract corresponding regional image features. We refer this procedure as *sparse feature sampling* since the bilinear interpolation takes only $\mathcal{O}(1)$ time for every sampling point. To obtain sampling points, a learnable linear layer is used to generate a set of relative sampling offsets for a token, conditioned on its embedding $\mathbf{t}$:

$$\{(\triangle x_i, \triangle y_i)\}_P = \text{Linear}(\mathbf{t}), \tag{2}$$

where the $i$-th sampling offset $(\Delta x_i, \Delta y_i)$ represents the relative position of sampling point $i$ with respect to a token RoI, and there are a total of $P$ sampling points for each token. The layer normalization (Ba et al., 2016) over $\mathbf{t}$ before the linear layer here, as well as linear layers below, is omitted for clarity. Then, these offsets are translated to absolute sampling locations $(\tilde{x}, \tilde{y})$ in an image with the RoI $\mathbf{b} = (x, y, w, h)$:

$$\begin{cases} \tilde{x}_i = x + 0.5 \cdot \triangle x_i \cdot w, \\ \tilde{y}_i = y + 0.5 \cdot \triangle y_i \cdot h, \end{cases} \tag{3}$$

for every $i$. To ensure stable training, a standard normalization is applied to $\{(\triangle x_i, \triangle y_i)\}_P$ using three standard deviations, which keeps a majority of sampling points within the RoI.

With sparse feature sampling, SparseFormer extracts image features by direct bilinear interpolation based on these explicit sampling points, eliminating the need for dense grid traversal. Given an input image $\mathbf{I} \in \mathbb{R}^{C \times H \times W}$ with $C$ channels, the shape of the sampled feature matrix $\mathbf{x}$ for a token is $\mathbb{R}^{P \times C}$. The computational complexity of this sparse feature sampling procedure is $\mathcal{O}(NPC)$ with the number of latent tokens $N$ given, independent of the input image size $H \times W$.

**Adaptive feature decoding.** With image features $\mathbf{x}$ sampled for a token, *how to effectively decode them to build a token representation* is another key question. A linear layer of $\mathbb{R}^{P \times C} \to \mathbb{R}^d$ can be the simplest method to embed features into a token embedding, but we find it rather ineffective. We use the adaptive mixing method (Gao et al., 2022) to decode sampled features to leverage spatial and channel semantics in an adaptive way. Specifically, we use a lightweight network $\mathcal{F} : \mathbb{R}^d \to \mathbb{R}^{C \times C + P \times P}$ whose input is the token embedding $\mathbf{t}$ to produce conditional channel decoding weight and spatial decoding weights $\mathbf{M}_c$ and $\mathbf{M}_s$ and decode sampled features $\mathbf{x}$ with these weights:

$$[\mathbf{M}_c | \mathbf{M}_s] = \mathcal{F}(\mathbf{t}), \mathbf{M}_c \in \mathbb{R}^{C \times C}, \mathbf{M}_s \in \mathbb{R}^{P \times P}, \tag{4}$$

$$\mathbf{x}^{(1)} = \text{GELU}(\mathbf{x}^{(0)} \mathbf{M}_c) \in \mathbb{R}^{P \times C}, \tag{5}$$

$$\mathbf{x}^{(2)} = \text{GELU}(\mathbf{M}_s \mathbf{x}^{(1)}) \in \mathbb{R}^{P \times C}, \tag{6}$$

where $\mathbf{x}^{(0)}$ is the sampled feature $\mathbf{x}$, $\mathbf{x}^{(2)}$ is the output of adaptive feature decoding, and GELU serves as activation function (Hendrycks & Gimpel, 2016). Our $\mathcal{F}$ choice is composed of two linear layers without activation functions, with the hidden dimension $d/4$ for efficiency. The final output is added back to the token embedding $\mathbf{t}$ by a linear layer. Adaptive feature decoding can be seen as a token-wise spatial-channel factorization of dynamic convolution (Jia et al., 2016), and it adds moderate convolutional inductive bias to SparseFormer. The adaptive feature decoding design allows SparseFormer to reason about what a token expects to see based on what the token has seen in stages before. Therefore, a token can also reason about *where to look* with RoI adjusting mechanism, which is described below.

**RoI adjusting mechanism.** A first quick glance at an image with human eyes is usually insufficient to fully understand what is in it. This is also the case for the focusing transformer. In the focusing transformer, a latent token RoI can be refined iteratively together with updating its embedding, where a token RoI $\mathbf{b} = (x, y, w, h)$ is adjusted to $(x', y', w', h')$ in a detection-like way (Ren et al., 2015):

$$x' = x + \Delta t_x \cdot w, \qquad y' = y + \Delta t_y \cdot h, \tag{7}$$

$$w' = w \cdot \exp(\Delta t_w), \quad h' = h \cdot \exp(\Delta t_h), \tag{8}$$

where $(\Delta t_x, \Delta t_y, \Delta t_w, \Delta t_h)$ are normalized adjusting deltas, generated by a linear layer whose input is the embedding in a token-wise way:

$$\{\Delta t_x, \Delta t_y, \Delta t_w, \Delta t_h\} = \text{Linear}(\mathbf{t}). \tag{9}$$

With the RoI adjusting mechanism and sufficient training, token RoIs can gradually focus on foregrounds after several iterations. It is worth noting that unlike object detectors, we do not rely on any localizing signals for supervision. The optimization for the RoI adjusting mechanism is end-to-end and accomplished by back-propagating gradients from sampling points in the bilinear interpolation. Although there may be noisy gradients due to local and limited sampling points in bilinear interpolation, the optimization direction for RoI adjusting is still non-trivial as shown in experiments.

**Focusing transformer** can be generally considered a transformer "decoder" architecture, where its cross attention to image features is modified to sparse feature sampling. Instead of directly traversing grid features in standard transformer decoders, the focusing transformer extracts image features sparsely, whose computation complexity is therefore independent of input resolution. We make the focusing transformer as lightweight as possible and make it one-block but repeating in several times with same parameters. Also, the token dimension in the focusing transformer is set the half of the cortex transformer. After the focusing transformer, only latent token embeddings are fed into the following transformer since token RoIs are not further used in the cortex transformer.

**Cortex transformer.** The cortex transformer follows a standard transformer encoder architecture except for the first block with the "cross attention" in the focusing transformer to decode features of large token dimension. The cortex transformer consists up of multiple independent blocks and take the most of parameters and computation in SparseFormer. This heavy cortex transformer processes visual signals from the focusing transformer, similar to how the brain cerebral cortex processes visual input from the eyes.

## 3.2 OVERALL SPARSEFORMER ARCHITECTURE

The overall architecture in SparseFormer is depicted in Figure 2. The image features are computed only once by lightweight convolution and shared across sampling stages. The final classification is done by averaging embeddings $\{\mathbf{t}_i\}_N$ over latent tokens and applying a linear classifier to it.

**Early convolution.** As discussed earlier, gradients with respect to sampling points by bilinear interpolation might be very noisy. The case can be particularly severe when handling raw RGB inputs, as the nearest four RGB values on grid are usually too noisy for accurate estimation of local gradients. To improve training stability, we incorporate early convolution, similar to vision transformers (Xiao et al., 2021), but make it as lightweight as possible.

**Sparsity of the architecture.** It is important to note that the number of latent tokens in SparseFormer is fixed and does not necessarily depend on the input resolution. The sparse feature sampling procedure extracts features from image feature maps also in a sparse and non-traversing way. The computational complexity and memory footprints in the latent space is therefore irrelevant to the

Table 1: **Configurations** of SparseFormer variants. FLOPs is with the input image size $224^2$.

| variant | #tokens $N$ | foc. dim $d_f$ | cort. dim $d_c$ | cort. stage $L_c$ | FLOPs | #params |
|---------|-------------|----------------|-----------------|-------------------|-------|---------|
| tiny (T) | 49 | 256 | 512 | 8 | 2.0G | 32M |
| small (S) | 64 | 320 | 640 | 8 | 3.8G | 48M |
| base (B) | 81 | 384 | 768 | 10 | 7.8G | 81M |

input size. Hence, it is reasonable to refer our method as a sparse visual architecture. Moreover, the latent space capacity of SparseFormer also exhibits sparsity. The maximum capacity of the latent space, $N \cdot d_c$, is $81 \cdot 768$ described below in Table 1, which is still smaller than the input image ($3 \cdot 224^2$). This also distinguishes SparseFormer from Perceivers (Jaegle et al., 2021; 2022), whose typical latent capacity ($512 \cdot 1024$) exceeds the input image size. Note that our presented approach differs a lot from post-training token reduction techniques (Rao et al., 2021; Liang et al., 2022; Fayyaz et al., 2022; Bolya et al., 2022; Yin et al., 2022). SparseFormer learns to represent an image using sparse tokens right from the start. In contrast, token reduction techniques are typically applied to pre-trained vision transformers. It is appealing to further reduce latent tokens in SparseFormer with these methods, but this is beyond the scope of this paper.

**Extension to video classification.** Video classification requires more intensive computing due to multiple frames. Fortunately, SparseFormer can be easily extended to video classification with minor extra efforts. Given a video feature $\mathbf{V} \in \mathbb{R}^{C \times T \times H \times W}$, the only problem is to deal with the extra temporal axis compared to the image feature $\mathbf{I} \in \mathbb{R}^{C \times H \times W}$. To address this, we associate the token RoI with an extension $(t, l)$, the center temporal coordinate $t$ and $l$ represents the temporal length to make RoI a tube. In the sparse feature sampling procedure, an extra linear layer is introduced to produce temporal offsets, and we transform them to 3D sampling points $\{(\tilde{x}_i, \tilde{y}_i, \tilde{z}_i)\}_P$. Here, the bilinear interpolation is replaced by the trilinear one for 4D input data. Similarly, the RoI adjusting is extended to the temporal dimension. Other operators such as early convolution, adaptive feature decoding, self-attention, and FFN remain untouched. For larger latent capacities, we inflate tokens along the temporal axis by a factor of $n_t$ and initialize their $(t, l)$ to cover all frames, where $n_t$ is typically smaller than the frame count $T$ (*e.g.*, 8 versus 32).

## 4 EXPERIMENTS

We benchmark our presented SparseFormers on the ImageNet-1K classification (Deng et al., 2009) and Kinetics-400 (Carreira & Zisserman, 2017) video classification. We also report our preliminary trials on down-streaming tasks, semantic segmentation and object detection, in the appendix.

**Model configurations.** We use ResNet-like early convolutional layer (a $7 \times 7$ stride-2 convolution, a ReLU, and a $3 \times 3$ stride-2 max pooling) to extract initial 96-d image features. We design Sparse-Former variants from 2G to ~8G FLOPs in Table 1. We mainly scale up the number of latent tokens $N$, the dimension of the focusing and cortex transformer $d_f$ and $d_c$, and blocks of the cortex transformer $L_c$. The number of the recurrence of the one-block focusing transformer, $L_f$, for all variants is set to 4. The number of latent tokens is scaled up as modestly as possible so that it is smaller

Table 2: Comparison of different architectures on ImageNet-1K classification. The input resolution is $224^2$. The throughput is measured with FP32 on a single V100 GPU following (Liu et al., 2021a).

| method | top-1 | FLOPs | #params | throughput (img/s) |
|--------|-------|-------|---------|--------------------|
| ResNet-50 (Wightman et al., 2021) | 80.4 | 4.1G | 26M | 1179 |
| ResNet-101 (Wightman et al., 2021) | 81.5 | 7.9G | 45M | 691 |
| DeiT-S (Touvron et al., 2021) | 79.8 | 4.6G | 22M | 983 |
| DeiT-B (Touvron et al., 2021) | 81.8 | 17.5G | 86M | 306 |
| Swin-T (Liu et al., 2021a) | 81.3 | 4.5G | 29M | 726 |
| Swin-S (Liu et al., 2021a) | 83.0 | 8.7G | 50M | 437 |
| Perceiver (Jaegle et al., 2021) | 78.0 | 707G | 45M | 17 |
| Perceiver IO (Jaegle et al., 2022) | 82.1 | 369G | 49M | 30 |
| SparseFormer-T | 81.0 | 2.0G | 32M | 1270 |
| SparseFormer-S | 82.0 | 3.8G | 48M | 898 |
| SparseFormer-B | 82.6 | 7.8G | 81M | 520 |

Table 3: **Comparison with video classification methods** on Kinetics-400. The GFLOPs is in the format of a single view × the number of views. "N/A" indicates the numbers are not available.

| method | top-1 | pre-train | #frames | GFLOPs | #params |
|---|---|---|---|---|---|
| NL I3D (Wang et al., 2018) | 77.3 | ImageNet-1K | 128 | 359×10×3 | 62M |
| SlowFast (Feichtenhofer et al., 2019) | 77.9 | - | 8+32 | 106×10×3 | 54M |
| TimeSFormer (Bertasius et al., 2021) | 75.8 | ImageNet-1K | 8 | 196×1×3 | 121M |
| Video Swin-T (Liu et al., 2021b) | 78.8 | ImageNet-1K | 32 | 88×4×3 | 28M |
| ViViT-B FE (Arnab et al., 2021) | 78.8 | ImageNet-21K | 32 | 284×4×3 | 115M |
| MViT-B (Fan et al., 2021) | 78.4 | - | 16 | 71×5×1 | 37M |
| VideoSparseFormer-T | 77.9 | ImageNet-1K | 32 | 22×4×3 | 31M |
| VideoSparseFormer-S | 79.1 | ImageNet-1K | 32 | 38×4×3 | 48M |
| VideoSparseFormer-B | 79.8 | ImageNet-21K | 32 | 74×4×3 | 81M |

than dense vision transformers. The center of latent token RoIs is initialized to a grid, and sampling points for a token are also initialized to a grid. For the sake of consistency in building blocks, the first cortex transformer block also performs RoI adjusting, feature sampling, and decoding. We do not inject any positional information into latent tokens. **Training recipes.** For ImageNet-1K classification (Deng et al., 2009), we train the proposed SparseFormer according to the recipe in (Liu et al., 2021a), which includes the training budget of 300 epochs, the AdamW optimizer (Loshchilov & Hutter, 2017) with an initial learning rate 0.001, the weight decay 0.05 and sorts of augmentation and regularization strategies. The input resolution is fixed to $224^2$. We add EMA (Polyak & Juditsky, 1992) to stabilize the training. The stochastic depth (*i.e.*, drop path) (Huang et al., 2016) rate is set to 0.2, 0.3, and 0.4 for SparseFormer-T, -S, and -B.

For ImageNet-21K pre-training (Deng et al., 2009), we use the subset, `winter 2021 release`, as suggested by (Ridnik et al., 2021). We follow the pre-training recipe in (Liu et al., 2021a) with 60 epochs, an initial learning rate $2 \times 10^{-3}$, weight decay 0.05, and drop path 0.1. After pre-training, we fine-tune models with a recipe of 30 epochs, an initial learning rate $2 \times 10^{-4}$ with cosine decay and weight decay $10^{-8}$.

For training on Kinetics-400 Carreira & Zisserman (2017), we use ImageNet pre-trained weights to initialize SparseFormers. Since our architecture is endurable to large input sizes, the number of input frames is set to $T = 32$. To be specific, 32 frames are sampled from the 128 consecutive frames with a stride of 4. We mildly inflate initial latent tokens by $n_t = 8$ times in the temporal axis to cover all input frames. Our model is optimized by AdamW (Loshchilov & Hutter, 2017) on 32 GPUs following the training recipe in (Fan et al., 2021). We train the model for 50 epochs with 5 linear warm-up epochs. The mini-batch size is 8 videos per GPU. The learning rate is set to $5 \times 10^{-4}$, and we adopt a cosine learning rate schedule (Loshchilov & Hutter, 2016). For evaluation, we apply a 12-view testing scheme (three 3 spatial crops and 4 temporal clips) as previous work (Liu et al., 2021b).

## 4.1 MAIN RESULTS

**ImageNet-1K classification.** We first benchmark SparseFormer on the ImageNet-1K classification and compare them to other well-established methods in Table 2. SparseFormer-T reaches 81.0 top-1 accuracy on par with the well-curated dense transformer Swin-T (Liu et al., 2021a), with less than half FLOPs of it (2.0G versus 4.5G) and 74% higher throughput (1270 versus 726). The small and base variants of SparseFormer, SparseFormer-S, and -B also maintain a good tradeoff between the

Table 4: **Scaling up** of SparseFormer-B. Except for the 1K entry, all follow first the same pre-training on ImageNet-21K ($224^2$ input, 81 tokens) and then individual fine-tuning on ImageNet-1K.

| variant | pre-training data | resolution | top-1 | FLOPs | throughput (img/s) |
|---|---|---|---|---|---|
| B | IN-1K | $224^2$ | 82.6 | 7.8G | 520 |
| B | IN-21K | $224^2$ | 83.6 | 7.8G | 520 |
| B | IN-21K | $384^2$ | 84.1 | 8.2G | 444 |
| B | IN-21K | $512^2$ | 84.0 | 8.6G | 419 |
| B, $N = 144 \uparrow$ | IN-21K | $384^2$ | 84.6 | 14.2G | 292 |
| B, $N = 196 \uparrow$ | IN-21K | $384^2$ | 84.8 | 19.4G | 221 |

| $N$ | 9 | 16 | 25 | 36 | 49 | 64 | 81 |
|---|---|---|---|---|---|---|---|
| top-1 | 74.5 | 77.4 | 79.3 | 80.1 | 81.0 | 81.4 | 81.9 |
| GFLOPs | 0.5 | 0.8 | 1.1 | 1.6 | 2.0 | 2.7 | 3.3 |

(a)

| method | SF | ViT/32 | ViT/32* | conv×4 | swin |
|---|---|---|---|---|---|
| top-1 | 81.0 | 72.8 | 74.3 | 79.4 | 79.7 |
| GFLOPs | 2.0 | 1.4 | 1.7 | 2.2 | 2.0 |

(b)

| $L_f$ | top-1 | GFLOPs |
|---|---|---|
| nil | 77.8 | 1.6 |
| 1 | 79.7 | 1.7 |
| 4 | 81.0 | 2.0 |
| 8 | 81.0 | 2.5 |

(c)

| $P$ | top-1 | GFLOPs |
|---|---|---|
| 16 | 80.3 | 1.9 |
| 36 | 81.0 | 2.0 |
| 64 | 81.3 | 2.3 |

(d)

| img feat. | top-1 | GFLOPs |
|---|---|---|
| RGB | fail | 1.5 |
| ViT/8-embed | 78.4 | 1.9 |
| early conv | 81.0 | 2.0 |
| ResNet C1+C2 | 82.2 | 3.1 |

(e)

| decode | top-1 | GFLOPs |
|---|---|---|
| linear | 78.5 | 1.9 |
| static, mix | 80.1 | 1.9 |
| adaptive, mix | 81.0 | 2.0 |

(f)

Table 5: **Ablation study** on SparseFormer: (a) the number of latent tokens $N$; (b) the method to extract 49 tokens other than the focusing transformer; (c) the repeats of the focusing transformer, $L_f$; (d) the number of sampling points for a token, $P$; (e) image features to sample and decode; (f) the decoding approch. The default choice for SparseFormer is colored  gray .

performance and actual throughput over highly-optimized CNNs or transformers. We can find that Perceiver architectures (Jaegle et al., 2021; 2022), which also adopt the latent transformer, incur extremely large FLOPs and have impractical inference speed due to a large number of tokens (*i.e.*, 512) and dense traversing cross attention.

**Scaling up SparseFormer.** We scale up the base SparseFormer variant in Table 4. We first adopt ImageNet-21K pretraining, and it brings 1.0 top-1 accuracy improvement. Then we investigate SparseFormer with large input resolution fine-tuning. Large resolution inputs benefit SparseFormer (0.5↑ for $384^2$) only extra 5% FLOPs. Moreover, we try a more aggressive way to scale up the model by reinitializing tokens (*i.e.*, embeddings and RoIs) with a more number *in fine-tuning*, and find it with better results. We leave further scaling up to future work.

**Kinetics-400 classification.** We investigate the extension of SparseFormer to the video classification task. Results on the Kinetics-400 dataset are reported in Table 3. Our VideoSparseFormer-T achieves the performance of well-established video CNNs (I3D or SlowFast) with a much lower computational burden. Surprisingly, our VideoSparseFormer-S pre-trained on ImageNet-1K even surpasses the ViT-based architectures pre-trained on ImageNet-21K, like TimeSFormer (Bertasius et al., 2021) and ViViT (Arnab et al., 2021). Furthermore, our VideoSparseFormer-S pre-trained on ImageNet-21K can improve the performance to 79.8 with only 74 GFLOPs.

## 4.2 ABLATION STUDY

We ablate key designs in SparseFormer-T on ImageNet-1K due to limited computational resources.

**The number of latent tokens**, $N$, is a crucial hyper-parameter in SparseFormer that controls the capacity of the latent space. The impact of different numbers of latent tokens on the performance is shown in Table 5a. The performance of SparseFormer is heavily influenced by $N$, and increasing $N$ to 81 enables SparseFormer-T to achieve similar performance as SparseFormer-S. Nevertheless, we are still in favor of fewer tokens for efficiency and sparsity.

**The focusing transformer.** We ablate the focusing transformer and the early convolution with direct token extraction methods, while keeping the cortex transformer unchanged, in Table 5b. The 'ViT/32' indicates using $32 \times 32$ patch as a token (Dosovitskiy et al., 2021) (49 in all) and the transformer configurations follow SparseFormer-T in Table 1. The 'ViT/32*' adds two more transformer blocks. The 'conv×4' approach uses four strided convolution, each doubling the channel, to produce $7 \times 7$ feature map as 49 tokens. The 'swin' method exploits 4 successive local shifted attention and patch merge (Liu et al., 2021a) to obtain 49 tokens. Note that in contrast to SparseFormer, all methods in Table 5b extract 49 tokens in a dense manner from the original image space. We also investigate the impact of iterations of the focusing transformer in Table 5c.

**Sparsity of sampling points.** The other important factor for the sparsity of the proposed method is the number of sampling points in the feature sampling. Table 5d shows the ablation on this. Compared to increasing the number of latent tokens (*e.g.*, $49 \rightarrow 64$, 30% up, 81.4 accuracy), more

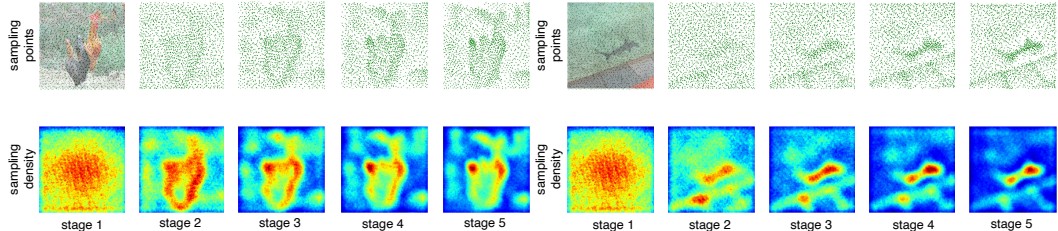

Figure 3: **Visualizations** of sampling points and their sampling density maps across sampling stages in SparseFormer-S. Stage 1-4 refer to the feature sampling in the focusing Transformer, and Stage 5 refers to the cortex Transformer. Better view with zoom-in.

sampling points are not economical for better performance, and 81.3 accuracy needs 77% more ($36 \rightarrow 64$) sampling points. We choose 36 sampling points as our default for efficiency.

**Image features and how to decode sampled features.** Table 5e investigates input image features to be sampled into the latent space in SparseFormer. As discussed before, input image features can be in the raw RGB format, but we find it rather hard to train[1]. We ablate the early convolution with ViT-like embedding layer (Dosovitskiy et al., 2021) and ResNet C1+C2 blocks (He et al., 2016) (the early convolution and the first three bottleneck blocks before downsampling). Table 5f ablates how to decode sampled features for a token. The static mixing uses the static weights, which are not conditional on token embedding, and performs worser than adaptive mixing weights.

**Inflation of latent tokens on video classification.** We also investigate the inflation rate of tokens on videos. Intuitively, video data with multiple frames need more latent tokens than a static image to model. Results in Table 6 show this. Note that the input video has 32 frames, but the token inflation rate 8 is already sufficient for the favorable performance of VideoSparseFormer. As a contrast, dense CNNs or Transformers usually require at least exactly #frames times the computational cost if no temporal reduction is adopted. This also validates the sparsity of the proposed SparseFormer method on videos.

Table 6: **Different inflation rates** of VideoSparseFormer-T on Kinetics-400.

| inflation | top-1 | GFLOPs |
|---|---|---|
| 1 | 69.5 | $7 \times 4 \times 3$ |
| 4 | 74.7 | $13 \times 4 \times 3$ |
| 8 | 77.9 | $22 \times 4 \times 3$ |
| 16 | 78.2 | $32 \times 4 \times 3$ |

### 4.3 VISUALIZATIONS

As discussed in Section 3, we argue that SparseFormer with the RoI adjusting mechanism and sparse feature sampling, can reason about where to look and focus on foregrounds. To show this, we perform visualizations of token sampling points across different sampling stages in Figure 3. We apply kernel density estimation (KDE (Rosenblatt, 1956)) spatially about sampling points with top-hat kernels to obtain the sampling density map. We can find that SparseFormer initially looks at the image in a relatively uniform way and gradually focuses on discriminative details of foregrounds. With *minimal classification supervision* (*i.e.*, no localizing signals), SparseFormer can roughly learn *where discriminative foregrounds are*.

## 5 CONCLUSION

In this paper, we have presented a new vision architecture, SparseFormer, to perform visual recognition with a limited number of tokens along with the transformer in the latent space. To imitate human perception behavior, we design SparseFormer to focus these sparse latent tokens on discriminative foregrounds and make a recognition sparsely. As a very initial step to the sparse visual architecture, SparseFormer consistently yields promising results on challenging image classification and video classification benchmarks with a good performance-throughput tradeoff. We hope our work can provide an alternative way and inspire further research about sparse visual understanding.

---

[1]Preliminary SparseFormer design using raw RGB input achieve approximately 60% top-1 accuracy.

**Acknowledgement**. This project is supported by the National Research Foundation, Singapore under its NRFF Award NRF-NRFF13-2021-0008.

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

## 6    APPENDIX

### A.1.    SPARSEFORMER ITSELF AS OBJECT DETECTOR

Since the SparseFormer architecture produces embedding and RoI together for a token given an image, it is natural to ask whether SparseFormer *per se* can perform object detection task? The answer is *yes*. In other words, we can train a SparseFormer model to detect objects *without making any architectural changes* by simply adding a final classifier and a RoI refining layer upon it.

Specifically, we follow the training strategy of DETR (Carion et al., 2020) to train a SparseFormer-S for object detection. We adopt the ImageNet-1K pre-trained SparseFormer-S in the main paper. We first inflate the number of latent tokens to $400$ by re-initializing token embeddings to the normalization distribution and the center of token RoIs to the uniform distribution on $[0, 1]$. The RoI height and width is $0.5 \times 0.5$ still. We use a fixed set of $100$ latent tokens to detect objects. The other tokens, which are not used for detection, aim to enrich the semantics in the latent space. We do not change the matching criterion together with the loss function of DETR and we train for 300 epochs. The final classifier is simply one-layer FC layer and the RoI refining layer is 2-layer FC layer, also following DETR. The final refining of RoIs is performed in the same way as RoI adjustment in the main paper. The result is shown in Table 7.

| detector | GFLOPs | AP | AP$_{50}$ | AP$_{75}$ | AP$_s$ | AP$_m$ | AP$_l$ |
|---|---|---|---|---|---|---|---|
| DETR | 86 | 42.0 | 62.4 | 44.2 | 20.5 | 45.8 | 61.1 |
| DETR-DC5 | 187 | 43.3 | 63.1 | 45.9 | 22.5 | 47.3 | 61.1 |
| SparseFormer-S | 27 | 26.4 | 43.8 | 26.6 | 8.3 | 26.0 | 45.0 |

Table 7: Detection performance of SparseFormer-S on MS COCO (Lin et al., 2014) `val` set.

Although the performance of SparseFormer is currently inferior to DETR, it is important to note that this is very preliminary result and we do not add any additional attention encoders or decoders to SparseFormer for object detection. Actually, SparseFormer can be considered a decoder-only architecture for object detection if we treat the early convolution part as the embedding layer.

### A.2.    SPARSEFORMER PERFORMING PER-PIXEL TASK

SparseFormer learns to represent an image by limited tokens in the latent space and outputs token embeddings with their corresponding RoIs. It is appealing to investigate whether SparseFormer can perform per-pixel tasks, like semantic segmentation. Yet, SparseFormer itself cannot perform dense tasks since it outputs discrete token set. However, we can *restore* a dense structured feature map from these discrete tokens by the vanilla cross-attention operator and build a final classifier upon the dense feature map.

Specifically, to perform semantic segmentation, we use a location-aware cross-attention operator to map latent token embeddings back to the structured feature map, whose height and width are one fourth of the input image (namely, stride 4, $H/4$ and $W/4$). The location-aware cross-attention is the vanilla cross-attention with geometric prior as biases in attention matrix:

$$\text{Attn}(Q_{ds}, K_{lt}, V_{lt}) = \text{Softmax}(Q_{ds}K_{lt}^T/\sqrt{d} + B)V_{lt},$$

where $Q_{ds} \in \mathbb{R}^{N_{ds} \times d}$ is the query matrix for the dense map ($N_{ds} = H/4 * W/4$), $K_{lt}, V_{lt} \in \mathbb{R}^{N \times d}$ are the key and value matrix for the latent tokens, respectively.

$$B_{i,j} = -(\frac{x_{ds,i} - x_{lt,j}}{w_{lt,j}})^2 - (\frac{y_{ds,i} - y_{lt,j}}{h_{lt,j}})^2$$

, where $(x_{lt}, y_{lt}, w_{lt}, h_{lt})$ is the RoI descriptor for a latent RoI and $(x_{ds}, y_{ds})$ are $x$ and $y$ coordinates for a unit on the dense feature map. In our current design, the input dense map to cross attend latent tokens is the early convolved feature map, which has the same height and width $H/4$ and $W/4$. We put two $3 \times 3$ convolution layers and a following classifier on the restored feature map following common pratice. The results are shown in Table 8.

| segmentor | GFLOPs | mIoU | mAcc |
|---|---|---|---|
| Swin-T (Liu et al., 2021a) + UperNet (Xiao et al., 2018) | 236 | 44.4 | 56.0 |
| SF-T w/ 49 tokens | 33 | 36.1 | 46.0 |
| SF-T w/ 256 tokens | 39 | 42.9 | 53.7 |
| SF-T w/ 400 tokens | 43 | 43.5 | 54.7 |

Table 8: Semantic segmentation performance of SparseFormer-T on Ade20K (Zhou et al., 2019) validation set. The GFLOPs are computed with $512 \times 512$ input resolution.

We also inflate the number of latent tokens in the semantic segmentation as we do in object detector for better performance. The performance of SparseFormer-T with 400 latent tokens is near the well-established Swin (Liu et al., 2021a) and UperNet (Xiao et al., 2018) but with merely $1/8$ Swin-T's GFLOPs. This validates that our proposed SparseFormer can perform per-pixel task and is suitable to handle high resolution input data with limited latent tokens.

### A.3. MORE INFERENCE DETAILS

We benchmark more throughput comparisons here on a more recent A5000 GPU in Table 9 with both FP32 and FP16. The proposed SparseFormer architectures deliver high throughput and takes lower memory footprints at inference with both FP32 and FP16 data type. The advantage is more evident with FP16 data type.

| method | top-1 | FLOPs | FP32 throughput | FP32 mem | FP16 throughput | FP16 mem |
|---|---|---|---|---|---|---|
| ResNet-50 (Wightman et al., 2021) | 80.4 | 4.1G | 1269 | 3010MB | 2119 | 2906MB |
| ResNet-101 (Wightman et al., 2021) | 81.5 | 7.9G | 797 | 3228MB | 1318 | 3121MB |
| DeiT-S (Touvron et al., 2021) | 79.8 | 4.6G | 977 | 536MB | 2732 | 430MB |
| DeiT-B (Touvron et al., 2021) | 81.8 | 17.5G | 305 | 1516MB | 1112 | 1470MB |
| Swin-T (Liu et al., 2021a) | 81.3 | 4.5G | 688 | 1471MB | 1748 | 1076MB |
| Swin-S (Liu et al., 2021a) | 83.0 | 8.7G | 396 | 1715MB | 1073 | 1285MB |
| SparseFormer-T | 81.0 | 2.0G | 1207 | 1146MB | 2925 | 699MB |
| SparseFormer-S | 82.0 | 3.8G | 824 | 1328MB | 2182 | 849MB |
| SparseFormer-B | 82.6 | 7.8G | 475 | 1711MB | 1285 | 1395MB |

Table 9: Benchmarking SparseFormers with other architectures on A5000 with batch size 32.

### A.4. MODEL INITIALIZATIONS IN DETAILS

We initialize all weights of linear layers in SparseFormer unless otherwise specified below to follow a truncated normalization distribution with a mean of $0$, a standard deviation of $0.02$, and a truncated threshold of $2$. Biases of these linear layers are initialized to zeros if existing.

Sampling points for every token are initialized in a grid-like shape ($6 \times 6$ for 36 sampling points by default) by zeroing weights of the linear layer to generate offsets and setting its bias using `meshgrid`. Alike, we initialize the center of initial token RoIs (as parameters of the model) to the grid-like (*e.g.*, $7 \times 7$ for 49 SF-Tiny variant) shape in the same way. The token's height and width are set to half of the image's height and width, which is expressed as $0.5 \times 0.5$. We also try other initializations for tokens' height and width in Table 10. For training stability, we also initialize adaptive decoding in SparseFormer following (Gao et al., 2022) with an initial Xavier decoding weight (Glorot & Bengio, 2010). This initialization makes the adaptive decoding behaves like unconditional convolution (weights not dependent on token embeddings) at the beginning of the training procedure.

For alternative ways for token height and width initializations, we can find that there is no significant difference between the 'half' and 'cell' initializations. We prefer the 'half' initialization as tokens can see more initially. However, setting all token RoIs to the whole image, the 'whole' initialization, is lagging before other initializations. We suspect that the model is unable to differentiate between different tokens and is causing training instability due to identical RoIs and sampling points for all tokens.

| width and height initialization | top-1 |
|---|---|
| half, $0.5 \times 0.5$ | 81.0 |
| cell, $1/\sqrt{N} \times 1/\sqrt{N}$ | 81.0 |
| whole, $1.0 \times 1.0$ | 80.2 |

Table 10: Alternative ways to initialize the token height and width for SparseFormer-T. $N$ is the number of latent tokens for the 'cell' initialization. The 'cell' initialization tiles RoIs without overlapping over the image. The 'whole' initialization is with all token RoIs centered at the image center.

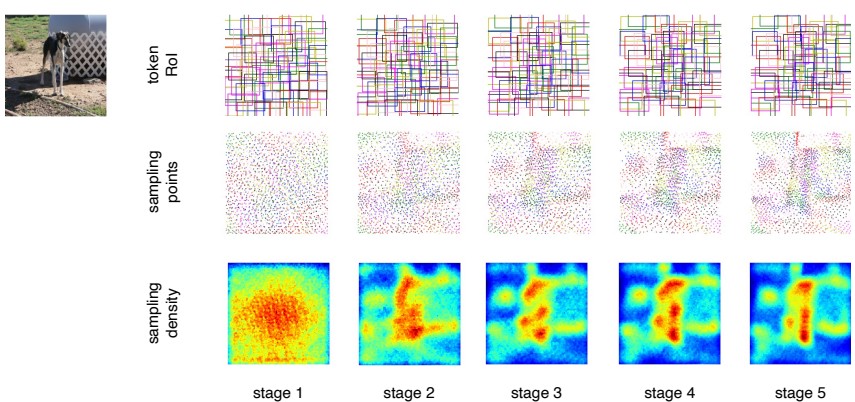

Figure 4: **More visualizations** of token RoIs, their sampling points, and density across sampling stages in SparseFormer-S ($64$ tokens). RoIs and sampling points of different tokens are colored with different colors. Better view with zoom-in.

### A.4. VISUALIZATIONS

**More visualizations on RoIs and sampling points.** In order to confirm the general ability of SparseFormer to focus on foregrounds, we present additional visualizations in Figure 4 and 5 with ImageNet-1K (Deng et al., 2009) validation set inputs. Note that these samples are not cherry-picked. We observe that SparseFormer progressively directs its attention towards the foreground, beginning from the roughly high contrasting areas and eventually towards more discriminative areas. The focal areas of SparseFormer adapt to variations in the image and mainly concentrate on discriminative foregrounds when the input changes. This also validate the semantic adaptability of SparseFormer to different images.

**Visualizations on specific latent tokens.** We also provide visualizations of specific latent tokens across stages to take a closer look at how the token RoI behaves at the token level. We choose 5 tokens per image that respond with the highest values to the ground truth category. To achieve this, we remove token embedding average pooling and place the classifier layer on individual tokens. The visualizations are shown in Figure 6. We can observe that the token RoIs progressively move towards the foreground and adjust their aspect ratios at a mild pace by stage.

**Visualizations on disturbed input images.** We also show visualizations on disturbed input images in Figure 7, where images are either random erased or heavily padding with zero values or reflection. We can see that although SparseFormer initially views the image in a almost uniform way, it learns to avoid sampling in uninformative areas in subsequent stages. This illustrates the robustness and adaptability of SparseFormer when dealing with perturbed input images.

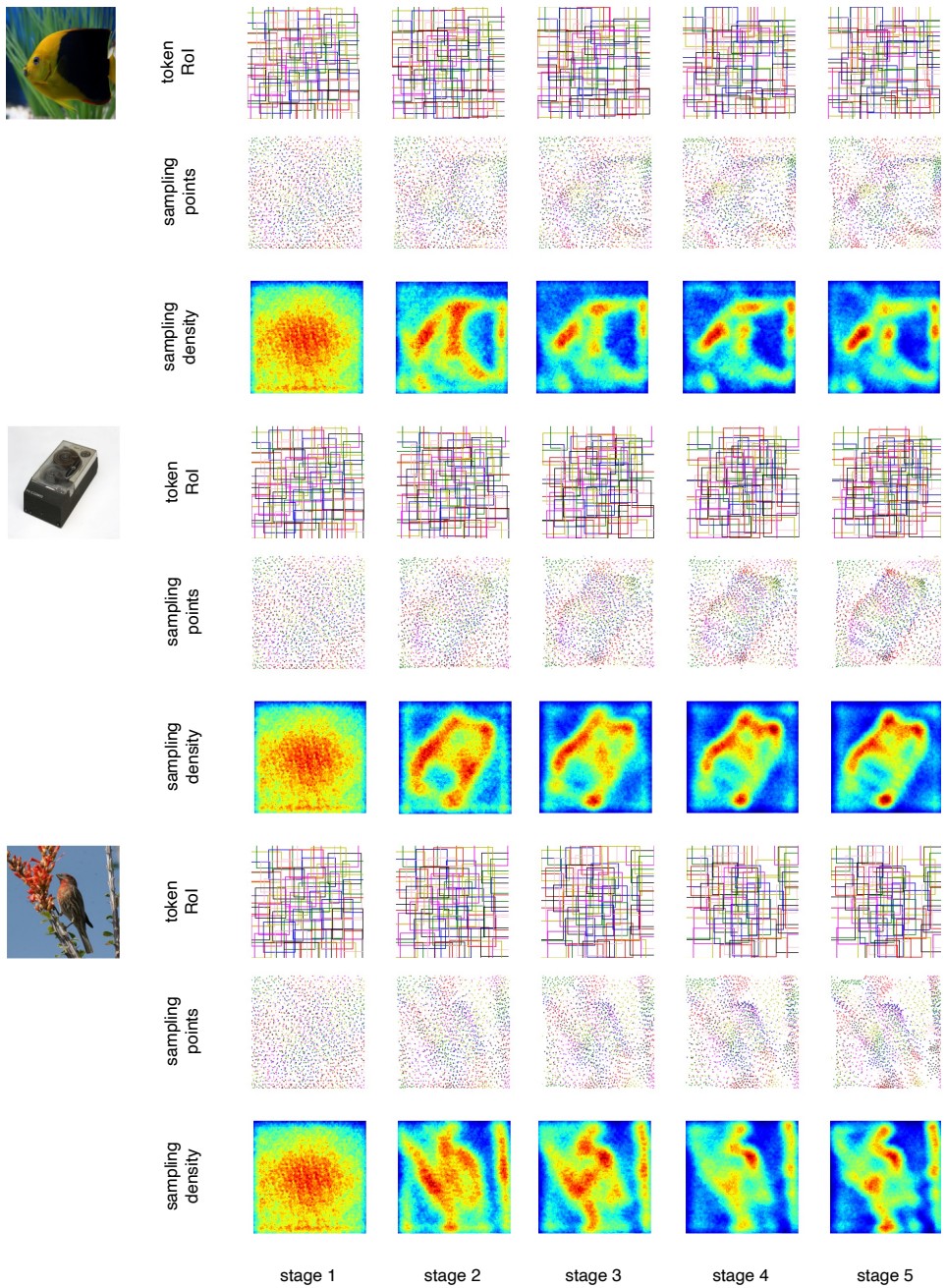

Figure 5: **More visualizations** (cont'd).

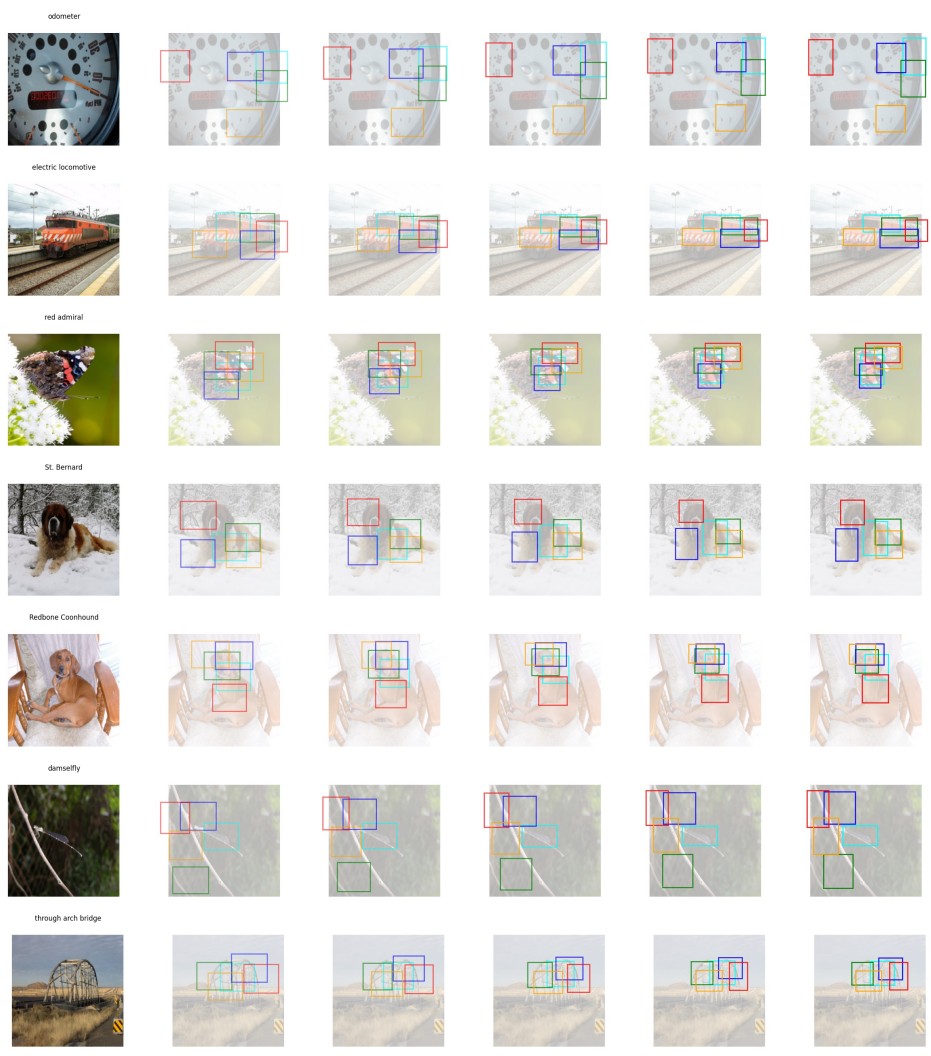

Figure 6: Visualizations on top-5 tokens responding to the ground-truth category across stages.

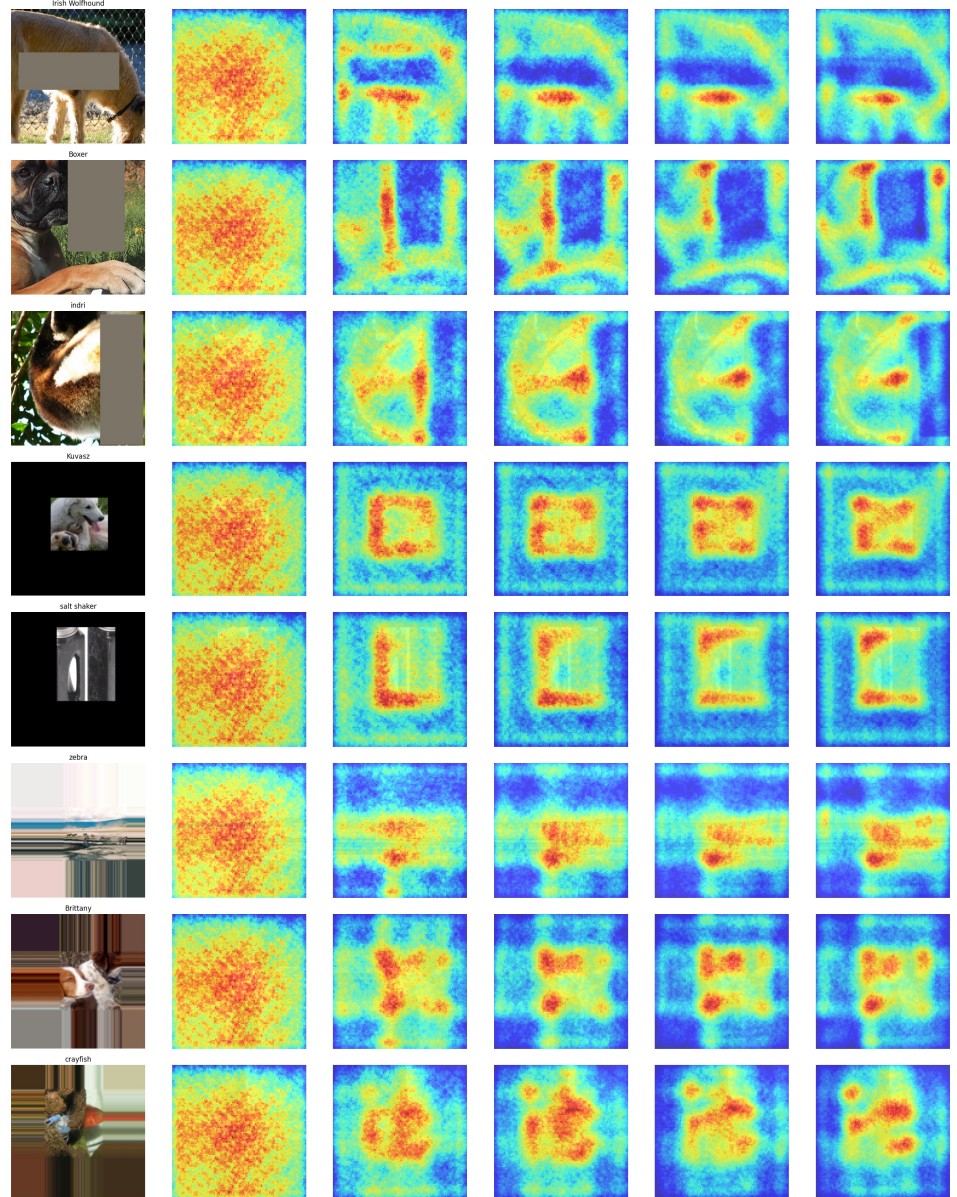

Figure 7: Visualizations on sampling density maps when disturbing input images.

