# OpenReview forum: "SparseFormer: Sparse Visual Recognition via Limited Latent Tokens"
_ICLR.cc/2024/Conference — ICLR 2024 poster_

### Official Review · Reviewer_H2So · 2023-10-15

**Soundness:** 3 good
**Presentation:** 3 good
**Contribution:** 2 fair
**Rating:** 6
**Confidence:** 5

**Summary:**

This paper introduces SparseFormer, an innovative vision transformer designed for efficiency, which encodes images into a select number of sparse tokens in a latent space. The efficacy and computational economy of SparseFormer are showcased through its performance in ImageNet and video classification tasks.

**Strengths:**

1. the SparseFormer model it introduces attains impressive results with a notable reduction in computational cost and latency, highlighting its efficiency and practicality in application.
2. it is composed with a clear presentation, making it accessible and understandable

**Weaknesses:**

The experimental validation does not appear to be solid, such as the detection results and the ablation. see details in Question part
The novelty of SparseFormer is somewhat constrained, as it does not substantially deviate from existing methods in the field.
There is an absence of comparative analysis with other efficiency-oriented techniques, such as token pruning

**Questions:**

Could you provide insight into why the detection results are not more favorable, especially considering that your Region of Interest (RoI) mechanism appears to be well-suited for detection tasks?
Additionally, the paper does not include an ablation study on adjusting the RoI mechanism, which leaves its importance in the proposed method somewhat ambiguous. Could you clarify the necessity of the RoI mechanism within your framework?

---

> ### Author Response · Authors · 2023-11-22
> **Response**
>
> __W1. Comparison with token pruning__
>
> Thank you for your valuable suggestion.
> We here compare SparseFormers with one of the most typical and effective token pruning method, ToMe [1], which can be used off-the-shelf to trained models. The throughput is measured on a single NVIDIA A5000 with batch size 32. Note that AugReg [2] models here are pre-trained on ImageNet-21K and we use them as strong baselines.
>
> |      model     |  acc | throughput | FLOPs |
> |:--------------:|:----:|:----------:|--------------|
> | DeiT-S         | 79.8 | 977        | 4.6G        |
> | DeiT-S, ToMe@13    | 78.5 | 1606        | 2.4G        |
> | DeiT-B         | 81.8 | 305       | 17.5G       |
> | DeiT-B, ToMe@13         | 80.0 | 481       | 10.4G       |
> |--------------|----|----------|--------------|
> | IN-21K pretrained AugReg-S         | 81.3 | 977        | 4.6G        |
> | IN-21K pretrained AugReg-S, ToMe@5   | 81.0 | 1065        | 3.7G        |
> | IN-21K pretrained AugReg-S, ToMe@13   | 79.3 | 1606        | 2.4G        |
> | IN-21K pretrained AugReg-B         | 84.6 | 305       | 17.5G       |
> | IN-21K pretrained AugReg-B, ToMe@13         | 82.5 | 481       | 10.4G       |
> |--------------|----|----------|--------------|
> | SparseFormer-T | 81.0 | 1207       | 2.0G       |
> | SparseFormer-S | 82.0 | 824        | 3.8G       |
> | SparseFormer-B | 82.6 | 475        | 7.8G       |
>
> The table shows that even with ImageNet-21K pre-training, the post-training ToMe with AugReg-B only just matches the same performance as SparseFormer-B, which was trained from scratch on ImageNet-1K.
> The smaller SparseFormer-T also offers a better trade-off over AugReg-S w/ ToMe.
>
> __W2. Why the detection results are not more favorable__
>
> This could be because we did not include any form of positional information in the latent token embeddings, as described in Section 4 model configurations. As a result, the attention between these token embeddings is not aware of their positional relationships.
> Possessing positional information is crucial for achieving optimal detection performance, as validated in the original DETR [3].
> The detection transformer pipeline relies on positional information for accurate localization and to suppress redundant detections by utilizing attention between tokens.
> Meanwhile, the segmentation is simple in sense that we can perform 'classification' on latent tokens that correspond to specific areas in an image, and map this 'classification' back into nearby pixels in the original image space using a simple operator (the location-aware cross-attention we used).
> Besides, using the exact same SparseFormer for the classification and its weights might be a reason, and we expect that introducing more newly initialized transformer blocks could alleviate this gap.

---

> ### Author Response · Authors · 2023-11-22
> **Response (cont'd)**
>
> __W3. Ablation study on the RoI adjusting mechanism__
>
> We have included the ablation study on the RoI adjuting mechanism from two aspects in Table 3 (b) and Table 3 (c) in the submission paper.
> We repeat these two tables here.
>
>
> | method    | SF   | ViT/32 | ViT/32* | conv×4 | swin |
> |-----------|------|--------|---------|--------|------|
> | top-1 acc | 81.0 | 72.8   | 74.3    | 79.4   | 79.7 |
> | GFLOPs    | 2.0  | 1.4    | 1.7     | 2.2    | 2.0  |
>
> Table 3 (b) shows the effectiveness of the focusing transformer to adjust RoIs and extract features to build 49 tokens, compared with the dense counterparts to produce 49 tokens.
> All entries in the table use the same cortex transformer configuration.
> The 'ViT/32' produces 49 tokens simply by patchifying an image into $32\times32$ patches, rather than using the focusing transformer.
> The 'ViT/32*' add two more transformer blocks to 'ViT/32'.
> The 'conv×4' uses 4 convolution+relu after the early convolution, where each convolution is with stride 2 and doubled output channel to produce 49 tokens.
> The 'swin' exploit the shifted local attention used in [4] to produce 49 tokens, similar to the 'conv×4'.
>
> The focusing transformer shows its importance to adjust RoIs to prioritize foregrounds and exclude backgrounds, and therefore enables SparseFormer to perform more accurate recognition with fewer latent tokens, compared with dense counterparts to produce tokens.
>
> | $L_f$ | top-1 | GFLOPs |
> |-------|-------|--------|
> | nil   | 77.8  | 1.6    |
> | 1     | 79.7  | 1.7    |
> | 4     | 81.0  | 2.0    |
> | 8     | 81.0  | 2.5    |
>
> Table 3 (c) studies the number of iteration of the focusing transformer to adjust RoIs, $L_f$.
> The 'nil' stands for the token RoIs are just learnable parameters of the model and do not adapt to different image content.
> In other words, the RoI for a token does not adjust at all and keeps the same for all images in the 'nil' entry, and as expected, it is with an inferior result.
> We can see that the iteration number is vital to the final performance, as the focusing transformer with the insufficient iteration may not effectively adjust tokens to foregrounds.
>
> [1] Daniel Bolya, Cheng-Yang Fu, Xiaoliang Dai, Peizhao Zhang, Christoph Feichtenhofer, and Judy Hoffman. Token merging: Your vit but faster. ICLR 2023.
>
> [2] Andreas Steiner, Alexander Kolesnikov, Xiaohua Zhai, Ross Wightman, Jakob Uszkoreit, and Lucas Beyer. How to train your vit? data, augmentation, and regularization in vision transformers. Trans. Mach. Learn. Res., 2022,
>
> [3] Nicolas Carion, Francisco Massa, Gabriel Synnaeve, Nicolas Usunier, Alexander Kirillov, and Sergey Zagoruyko. End-to-end object detection with transformers. ECCV, 2020.
>
> [4] Ze Liu, Yutong Lin, Yue Cao, Han Hu, Yixuan Wei, Zheng Zhang, Stephen Lin, and Baining Guo. Swin transformer: Hierarchical vision transformer using shifted windows. CVPR, 2021

---

### Official Review · Reviewer_fz8s · 2023-10-29

**Soundness:** 3 good
**Presentation:** 3 good
**Contribution:** 2 fair
**Rating:** 6
**Confidence:** 4

**Summary:**

The paper introduces the SparseFormer, which modifies the standard Transformer model by using a small number of tokens in latent space to reduce its size and computational complexity.

**Strengths:**

1. Provides an alternative sparse paradigm ($i.e.,$) for vision modeling compared to existing Transformers. Reduces computation by operating on limited tokens.
2. Token ROI adjustment mechanism is effective at focusing on foregrounds.
3. Visualizations show the model progressively focuses on discriminative regions.

**Weaknesses:**

1. While the paper demonstrates the effectiveness of SparseFormer on classification tasks. The reviewer has concerns about the generalization to more complex scenarios. Appendix A.1 also points out the inferior performance compared to the recent transformer network. The use of specific sparse attention patterns might limit the model's ability to capture certain types of long-range dependencies in the images for downstream tasks.
2. In addition, the reviewer also has concerns about token ROI. Adjusting token ROIs lacks strong spatial supervision. Performance on dense prediction tasks ($i.e.,$ segmentation tasks) requiring precise localization may suffer. With complex images, the signal will be weak and may not focus on the meaningful pixels.

**Questions:**

Overall, this paper presents a step towards sparse vision architectures by a novel token ROI approach. The reviewer has no further questions, please see the weakness part.

**Details Of Ethics Concerns:**

No ethics concern.

---

> ### Author Response · Authors · 2023-11-22
>
> Thank you for reviewing our paper. The concerns are addressed in the following:
>
> __W1. Results in Appendix.1 and generalization to more complex scenarios__
>
> We admit that the detection performance of SparseFormer is lagging behind DETRs.
> We suspect that this is because we do not inject any positional information into latent tokens while the detection transformer training pipeline may require strong positional encodings to localize and suppress nearby redudant detections.
> Besides, we directly use the SparseFormer architecture for classification (without the introduction of extra transformer blocks) and its pre-trained weights to perform object detection, and this may not be the optimal design.
>
> We agree that the sparse attention patterns might limit the long-term and precise positional dependency, which is important to the object detection task.
> However, this does not mean that SparseFormer cannot handle complex scenarios.
> As a fact, the SparseFormer segmentation model performs well on much more complex ADE20K scenarios in Appendix. 2.
> This is because SparseFormer follows a spatially divide-and-conquer manner when dealing such scenarios, where the location-aware cross-attention operator enables a token to only respond to a specific area.
>
> __W2. Concerns about RoI localization on dense prediction tasks__
>
> We believe that the SparseFormer does not require the strong explicit spatial RoI localization supervision on dense prediction tasks.
> Indeed, for segmentation tasks, the spatial distribution of classification labels can serve as a coarse localization supervision.
> A token RoI in SparseFormer can be adjusted to focus on a consistent label area by such spatial label distributions and the end-to-end location-aware cross-attention discussed above.
> Besides that, the segmentation task does not necessitate well-localized token RoIs to perform well since the final classification is performed on the dense map, which is projected back from latent tokens and one pixel in the dense map is a mixture of nearby latent tokens with the semantic similarity.

---

### Official Review · Reviewer_fS4e · 2023-10-30

**Soundness:** 3 good
**Presentation:** 3 good
**Contribution:** 2 fair
**Rating:** 5
**Confidence:** 4

**Summary:**

The authors introduce SparseFormer, which comprises two main components: the Focusing Transformer and the Cortex Transformer. The Focusing Transformer addresses the challenge of extracting image features sparsely, decoding them into latent tokens, and adjusting token regions of interest (RoIs). The Focusing Transformer efficiently extracts image features with a computational complexity of O(N·P·C), where N is the number of latent tokens, regardless of the input image size. Evaluated on ImageNet, the authors demonstrated that the proposed method achieved 1.7x faster inference speed compared with Swin-T with small accuracy degradation. It also outperforms ResNet50 with a faster speed.

**Strengths:**

1. This paper is thoroughly motivated and exceptionally well-written. The concept of sparsifying input tokens holds paramount importance for vision transformers (ViTs) owing to the quadratic complexity with respect to sequence length in multi-head self-attention.

2. The authors have designed a functional solution, known as FocusTransformer, which improves upon the Perceiver method by introducing and dynamically adjusting regions of interest (RoIs). Experimental results compellingly demonstrate the effectiveness of this architecture on the ImageNet dataset.

3. The authors have not only illustrated how SparseFormer can reduce computational workload (measured in FLOPs), but they have also empirically shown a significant speedup on a V100 GPU under FP32 precision, further showcasing the efficacy of their proposed approach.

**Weaknesses:**

1. While the authors have put considerable effort into elucidating the disparities between SparseFormer and Perceiver, it remains challenging for me to find a fundamental difference between these two methodologies. In my estimation, the primary distinction appears to be the introduction of the FocusTransformer. However, upon examination of this architecture, I have also observed a clear similarity to DeformableDETR. Consequently, I find it challenging to pinpoint the truly innovative contributions of this paper.

2. The scope of the evaluation in this work appears somewhat limited. The presentation exclusively reports image classification results. However, Vision Transformers (ViTs) have showcased their efficacy across a diverse range of computer vision tasks, including object detection, semantic segmentation, and image generation. A majority of these applications typically demand high-resolution inputs, which makes the efficiency of reducing the number of visual tokens even more critical. My particular interest lies in understanding the applicability of the proposed approach to dense prediction tasks such as segmentation and image generation with diffusion models, given that the FocusTransformer seems to introduce token-level information loss.

3. The section on speed evaluation is extensive, but it may benefit from further solidity. The reliance on the V100 GPU, which is considered somewhat outdated, raises questions in the context of contemporary Deep Neural Network (DNN) inference, where there is a preference for using lower precision formats like INT8 and FP16 with a TensorRT backend. Even though the proposed architecture is light in terms of FLOPs, I am concerned about the potential efficiency of the DeformableDETR-like FocusTransformer when integrated with TensorRT. It would be great if the authors could provide relevant results in this regard.

**Questions:**

Please respond to my questions and concerns in "Weaknesses".

---

> ### Author Response · Authors · 2023-11-22
> **Response**
>
> Thank you for your reviewing our paper. We will address your concerns in the following:
>
> __W1. Similarity to Perceiver and DeformableDETR architectures__
> First, we would like to emphasize again that the aim of SparseFormer is to build a _sparse_ architecture for visual recognition tasks, and many efforts of this paper are put on _how to perform visual recognition with much fewer tokens in a transformer-like architecture_.
> This distinguishes SparseFormer from either Perceiver or DeformableDETR, where both latters do not attempt to reduce the computation.
>
> __Difference with Perceivers__
>
> We agree with that the primary distinction lies in the focusing transformer since
> its introduction enables SparseFormer to perform accurate recognition with a highly limited number of tokens.
> From Table 3(b) in the paper, where illustrate how 49 tokens are produced, one can see the effectiveness of the focusing transformer comparing with dense counterparts.
>
> As noted in the paper, Perceivers use a naive cross attention layer to traverse the pixel or feature space in a dense manner $\mathcal{O}(H\cdot W\cdot C)$, where the focusing transformer in our SparseFormer extract a few tokens from an image in a sparse way $\mathcal{O}(N\cdot P\cdot C)$, regardless of the input height and width. Also, the size of latent space is also sparse, the typical Perceiver use 512 tokens with 1024 dimension, resulting in a total 512\*1024 capacity, which is more than $8\times$ the SparseFormer-B uses (81 tokens with 768 dimension).
>
> __Difference with DeformableDETR__
>
> Building a sparse architecture necessitates a sparse variant of the basic operators on the image space.
> We must admit that DeformableDETR and SparseFormer share the similar idea to use bilinear interpolation to extract image features, but the motivation and details differs a lot:
> DeformableDETR aims to achieve the shift invariance in detection transformers (DETRs), where the naive cross and self attention layers in DETRs struggle to handle, while SparseFormer uses bilinear interpolation to efficiently sample image features.
> Also, deformable attention in DeformableDETR is typically performed on _the highly semantic feature map (e.g., after ResNets)_ and _in a multi-scale manner_, while SparseFormer uses _a single-level early conved feature map_ to perform sampling _in the very beginning of the network_.
> From this view, SparseFormer can be seen as a _decoder-only_ architecture for visual modeling except for the early convolution, which much differs from DeformableDETR that requires deformable self attention encoders on dense units upon the backbone.
>
> __W2. Dense prediction tasks and information loss__
>
> SparseFormer can perform dense prediction tasks like object detection and semantic segmentation.
> We have included experimental results and discuss them in Appendix A.1 and Appendix A.2.
> Specifically, Table 7 in the Appendix shows the detection results on MS COCO on a simple SparseFormer-S with only an extra linear classifier head and an extra MLP for regression, and the training recipe and loss also following the DETR.
> Given that no positional information is injected into latent tokens and no multi-scale features are used, SparseFormer can still achieve meaningful detection results with about $0.3\times$ FLOPs.
> Table 8 in the Appendix illustrates the segmentation performance of SparseFormer on ADE20K.
> One can see that SparseFormer reaches the result on par with the dense Swin-T w/ UperNet with $0.18\times$ GFLOPs.
>
> We deeply understand your concern about information loss introduced by SparseFormer since the latent token capacity (e.g., 81\*768) is indeed much smaller than the image size (e.g., 3\*224\*224).
> However, we believe that this is _not_ a significant issue for visual recognition tasks. Natural images contain a lot of pixel-level redundancy, such as backgrounds, which contributes little the final recognition.
> In fact, our sparse paradigm is exactly based on this redundancy and the proposed SparseFormer learns how to discard the redudancy, or how to perform 'information loss', with the recognition supervision.
> As for generation tasks like with diffusion models, this information loss issue becomes somewhat critical, and it is appealing to investigate how to incorporate SparseFormers into generation models with the least information loss.
> However, this is beyond the scope of this paper.

---

> > ### Author Response · Authors · 2023-11-22
> > **Response (cont'd)**
> >
> > __W3. More speed evaluation__
> >
> > Thank you for the advice.
> > We also benchmark more throughput comparisons here on a more recent NVIDIA A5000 with batch size 32.
> >
> > **NVIDIA A5000 with FP16 data type**
> > |      model     |  acc | throughput | inferece mem |
> > |:--------------:|:----:|:----------:|--------------|
> > | ResNet50       | 80.4 | 2119       | 2906MB       |
> > | ResNet100      | 81.5 | 1318       | 3121MB       |
> > | Swin-T         | 81.3 | 1748       | 1076MB       |
> > | Swin-S         | 83.0 | 1073       | 1285MB       |
> > | DeiT-S         | 79.8 | 2732       | 430MB        |
> > | DeiT-B         | 81.8 | 1112       | 1470MB       |
> > | SparseFormer-T | 81.0 | 2925       | 699MB        |
> > | SparseFormer-S | 82.0 | 2182       | 849MB        |
> > | SparseFormer-B | 82.6 | 1285       | 1395MB       |
> >
> >
> > **NVIDIA A5000 with FP32 data type**
> > |      model     |  acc | throughput | inferece mem |
> > |:--------------:|:----:|:----------:|--------------|
> > | ResNet50       | 80.4 | 1269       | 3010MB       |
> > | ResNet100      | 81.5 | 797        | 3228MB       |
> > | Swin-T         | 81.3 | 688        | 1471MB       |
> > | Swin-S         | 83.0 | 396        | 1715MB       |
> > | DeiT-S         | 79.8 | 977        | 536MB        |
> > | DeiT-B         | 81.8 | 305       | 1516MB       |
> > | SparseFormer-T | 81.0 | 1207       | 1146MB       |
> > | SparseFormer-S | 82.0 | 824        | 1328MB       |
> > | SparseFormer-B | 82.6 | 475        | 1711MB       |
> >
> > **NVIDIA A5000 Torch TensorRT FP32 default settings**
> > |      model     |  acc | throughput |
> > |:--------------:|:----:|:----------:|
> > | ResNet50       | 80.4 | 3180       |
> > | ResNet100      | 81.5 | 1851       |
> > | Swin-T         | 81.3 | 1480       |
> > | Swin-S         | 83.0 | 875        |
> > | DeiT-S         | 79.8 | 1840       |
> > | DeiT-B         | 81.8 | 676        |
> > | SparseFormer-T | 81.0 | 2835       |
> > | SparseFormer-S | 82.0 | 1565       |
> > | SparseFormer-B | 82.6 | 1019       |
> >
> > (We did try to benchmark with Torch TensorRT and FP16 data type but always encounter bugs dealing with type conversion.)
> >
> > From tables above, we can see that SparseFormer still enjoys the better tradeoff between accuracy and throughput in both FP16 and FP32 data type, along with the reduced inference memory.
> > The same conclusion can also be observed in the TensorRT inference.
> > Therefore, we believe that the focusing transformer would not be a factor to harm the overall efficiency with a TensorRT backend even in INT8 or FP16 setting.

---

> > > ### Comment · Reviewer_fS4e · 2023-11-22
> > >
> > > Thank you for providing additional insights. I am pleased to note that many of my concerns regarding latency have been effectively addressed, and as a result, I am inclined to update my score to 5. However, I would like to emphasize the importance of incorporating FP16/INT8 TRT results in future revisions.
> > >
> > > Concerning the novelty aspect, I believe that the core innovation of DeformableDETR lies in indexing the feature map based on query-associated coordinates. The choice between bilinear interpolation and deformable attention appears to yield minimal differences in essence. The specific feature maps indexed (lower-level v.s. higher-level, single-scale v.s. multi-scale) do not matter in my opinion. Besides, there is an existing follow-up paper of DeformableDETR, DETR3D that also adopts bilinear interpolation.
> > >
> > > Furthermore, to enhance the impact of the paper, I suggest considering the inclusion of more segmentation/detection results within the main body rather than relegating them to the appendix. The findings in Table 8 are promising, and I encourage the authors to provide additional details, such as results on diverse datasets, latency comparisons, and exploration of various model architectures, in their forthcoming submissions. This could significantly strengthen the comprehensiveness and applicability of their work.

---

> ### Author Response · Authors · 2023-11-23
>
> We greatly appreciate for your kind reassessment on our work. Till now, we have not managed to deal with the type conversion bug (the issue is already mentioned by others due to customized ops, but not solved, https://github.com/pytorch/TensorRT/issues/2113). We will report FP16/INT8 in future revisions once this bug is resolved.
>
> Regarding the novelty, we agree that our method follows the same idea of indexing feature maps based on query-associated coordinates with Deformable-DETR and its follow-ups.
> But the important thing is, that we show this query coordinate feature map indexing scheme can lead to sparse vision transformers with much fewer tokens, which is not explored at all to our best knowledge.
> Therefore, we believe that our SparseFormer per se has it novelty in exploring sparse vision architectures.
>
> We are extremely grateful for your valuable and detailed suggestions on how to enhance the impact, the comprehensiveness, and the applicability of our work, especially for results in Table 8.
> In future versions, we will reorganize this paper and cover as much experiments as possible per your suggestions.

---

### Official Review · Reviewer_SBDj · 2023-10-31

**Soundness:** 3 good
**Presentation:** 3 good
**Contribution:** 3 good
**Rating:** 8
**Confidence:** 3

**Summary:**

This paper proposed a sparse paradigm for visual recognition, and introduced a novel backbone named SparseFormer, which has a lower memory footprint and higher throughput compared to dense architectures, especially in the low-compute region.
Experiments show that the proposed method achieves a low memory and time cost while maintaining high performance.

**Strengths:**

1. The proposed SparseFormer is novel and solid.
2. While maintain the performance, SparseFormer has a low memory footprint and high throughout.
3. The experiments are solid.

**Weaknesses:**

None

**Questions:**

None

---

> ### Author Response · Authors · 2023-11-22
>
> We greatly appreciate your positive rating of our paper. Thank you for your reviewing and assessment.

---

### Meta-Review · Area_Chair_mryY · 2023-12-05

**Metareview:**

This submission received three positive scores and one negative score. The novelty of the proposed latent tokens has been conformed by reviewers. After reading the paper, the review comments and the rebuttal, the AC think the major concerns are about the presentation and experimental details, which have been mostly addressed by the rebuttal. The authors are encouraged to give a more clear presentation of the novelty and the FP16/INT8 TRT results in the camera-ready paper.

**Justification For Why Not Higher Score:**

N/A

**Justification For Why Not Lower Score:**

N/A

---

### Decision · Program_Chairs · 2024-01-16

Accept (poster)